# Overexpression of *IlHMA2*, from *Iris lactea*, Improves the Accumulation of and Tolerance to Cadmium in Tobacco

**DOI:** 10.3390/plants12193460

**Published:** 2023-09-30

**Authors:** Cui Li, Qinghai Wang, Xincun Hou, Chunqiao Zhao, Qiang Guo

**Affiliations:** Institute of Grassland, Flowers and Ecology, Beijing Academy of Agriculture and Forestry Sciences, Beijing 100097, China; licui@baafs.net.cn (C.L.); qinghaiw@sina.com (Q.W.); houxincun@baafs.net.cn (X.H.); zhaochunqiao@baafs.net.cn (C.Z.)

**Keywords:** *Iris lactea*, antioxidant enzymes, Cd translocation, phytoremediation

## Abstract

Long-distance transport cadmium (Cd) from roots to shoots is a key factor for Cd phytoremediation. Our previous study indicated that heavy metal P_1B2_-ATPases, IlHMA2, was involved in improving the accumulation of Cd via mediated long-distance transport Cd, contributing to the phytoremediation in Cd accumulator *Iris lactea*. However, whether the overexpression of *IlHMA2* could enhance the accumulation and tolerance to Cd remains unclear in plants. Here, we generated transgenic tobacco overexpressing *IlHMA2* and tested its effect on the translocation and accumulation of Cd and zinc (Zn), as well as the physio-biochemical characteristics under 50 mg/L Cd exposure. The overexpression of *IlHMA2* significantly increased Cd concentrations in xylem saps, resulting in enhanced root-to-shoot Cd translocation compared with wild-type. Meanwhile, overexpressing *IlHMA2* promoted Zn accumulations, accompanied by elevating proline contents and antioxidant enzyme activity (SOD, POD, and CAT) to diminish the overproduction of ROS in transgenic tobacco. These pieces of evidence suggested that higher Zn concentrations and lower ROS levels could tremendously alleviate Cd toxicity for transgenic tobacco, thereby improving the growth and tolerance. Overall, the overexpression of *IlHMA2* could facilitate Cd accumulation and enhance its tolerance in tobacco exposed to Cd contaminations. This would provide a valuable reference for improving Cd phytoremediation efficiency.

## 1. Introduction

Heavy metal pollutions have become serious environmental problems in the world [1]. Cadmium (Cd) is a non-essential heavy metal and easily enters into food chain to threaten human health [2]. Consequently, it is urgent to determine how to lower available Cd in soils. Phytoremediation is cost-effective and an environmentally friendly method to remediate soils contaminated with Cd, compared to the physical and chemical methods [3].

The long-distance transport of Cd from roots to shoots is thought to be a key hub for improving Cd phytoremediation efficiency [4]. Metal-transporting proteins play a crucial role in the translocation of heavy metals [5]. Heavy metal P_1B_-ATPases (HMAs) are involved in heavy metal transport or detoxification across biological membranes using the energy provided by ATP hydrolysis [6], whilst P_1B2_-ATPases HMA2 and HMA4 are plasma membrane zinc (Zn)/Cd transporters, which act as a pump, loading metals into the xylem [7]. The knockout of *AtHMA4* reduced Cd concentrations dramatically in shoots, indicating that it is a key player in roots-to-shoots Cd translocation [8]. However, functional redundancy is known to exist between HMA2 and HMA4 [9]. AtHMA2 is responsible for Zn efflux from root cells, which is required for cytoplasmic Zn homeostasis [10]. In rice, OsHMA2 localized at the plasma membrane of root pericycle cells and participated in the root-to-shoot Zn and Cd translocation [11]. Subsequently, Mills et al. (2012) [12] found that overexpressing plasma membrane P_1B2_-ATPases Zn transporter, *HvHMA2*, from barley could complement the shoot Zn-deficiency phenotype of the *hma2hma4* Arabidopsis double-mutant. Meanwhile, overexpressing wheat *TaHMA2* enhanced root-to-shoot Zn/Cd translocation in rice, but its Cd resistance was decreased under high Cd levels [13]. Similarly, enhanced Cd sensitivity was reported in tobacco plants expressing *Arabidopsis halleri AhHMA4* [14]. Interestingly, the overexpression of *AtHMA4* improved the root growth and root-to-shoot translocation of Zn and Cd, thus conferring increased Cd resistance [15]. Moreover, the heterogeneous expression of cucumber *CsHMA4* significantly enhanced Cd tolerance in yeast [16]. Taken together, these pieces of evidence suggested that HMA2/HMA4 is not only required for the maintenance of Zn homeostasis, but is also an important component to determine Cd phytoremediation efficiency. Notably, overexpressing stress response membrane protein genes, e.g., *OsSMP1*, significantly improved plant tolerance to Cd stress [17]. Indeed, the transfer of genes involved in any of these processes into fast-growing, high-biomass crops may improve their remediation potential [18]. However, whether the overexpression of *HMA2*/*HMA4* could increase the phytoremediation efficiency and Cd tolerance remains controversial in transgenic plants. 

Our previous research showed that plasma membrane transporter IlHMA2, from a Cd accumulator *Iris lactea* Pall. var. chinensis (Fisch.) Koidz., plays a crucial role in the root-to-shoot Cd translocation and the maintenance of Zn homeostasis via modulating Zn transport systems, thereby improving Cd tolerance [19]. However, little information is yet known about the contribution that overexpressing *IlHMA2* has in improving Cd phytoremediation efficiency in transgenic plants. In the current study, we generated transgenic tobacco overexpressing *IlHMA2* and investigated its translocation and accumulation of Cd and Zn, as well as the physio-biochemical characteristics under 50 mg/L Cd exposure. Our findings demonstrate that the overexpression of *IlHMA2* enhances the root-to-shoot translocation of Cd. In addition, a higher concentration of Zn and lower levels of ROS could alleviate Cd toxicity for transgenic tobacco, thereby improving its growth and tolerance. Therefore, it is a good idea to improve Cd phytoremediation efficiency using gene engineering approach.

## 2. Results

### 2.1. Overexpression of IlHMA2 Improved the Growth of Tobacco under Cd Exposure

To gain insight into the function of *IlHMA2* in tobacco, 35S:IlHMA2 was introduced into tobacco using *Agrobacterium*-mediated transformation. Twelve independent transgenic lines (OE1–OE12) were screened by molecular identification. The relative expression level of *IlHMA2* was detected in twelve transgenic plants by sqRT-PCR (Figure 1B). For example, the OE2 and OE11 line showed the highest and lowest expression level, respectively (Figure 1B). Thus, we selected the transgenic lines of OE2 and OE11 for following analysis.

To identify the response of transgenic tobaccos to Cd stress, both the WT and transgenic lines (OE2 and OE11) were treated with 50 mg/L Cd. As shown in Figure 1A, both WT and transgenic tobacco grew well under normal conditions. However, the growth of WT was severely inhibited and appeared to wilt under 50 mg/L Cd treatment, but transgenic lines did not show the symptoms of Cd toxicity. For example, shoot dry weight, root dry weight, plant height, and root length of the transgenic tobacco in OE2 is 1.52-, 1.87-, 1.65-, and 1.52-fold of WT under 50 mg/L Cd treatment, respectively (Figure 1C–F). These results indicated the growth parameter of the transgenic plants was significantly higher than that of WT under Cd exposure, suggesting that the overexpression of *IlHMA2* could improve the growth of transgenic tobacco.

### 2.2. Overexpression of IlHMA2 Reduced the Lipid Peroxidation and Overproduction of ROS in Tobacco under Cd Exposure

To investigate Cd-induced oxidative stress, contents of H_2_O_2_, MDA, and proline were examined in transgenic lines (OE2 and OE11). H_2_O_2_ content was maintained at the same level in WT and transgenic plants under normal growth conditions (Table 1). However, the level of H_2_O_2_ in OE2 and OE11 decreased by 52.8% and 42.6% compared to WT under Cd stress (Table 1). Similarly, there was no significant difference in MDA content between WT and transgenic plants under normal growth conditions, whilst it was decreased by 42.9% and 43.5% in OE2 and OE11 relative to WT under Cd treatment, respectively (Table 1). Meanwhile, under Cd treatment, the proline content in transgenic plants OE2 and OE11 was 2.0 and 2.1 times of WT plants (Table 1).

We further measured the antioxidant enzyme activity. The activity of superoxide dismutase (SOD), peroxidases (POD), or catalase (CAT) has no prominent difference between transgenic tobacco and WT plants in the absent of Cd condition. However, the activity of SOD and POD in transgenic tobacco was significantly increased. For example, the activity of SOD increased by 41.6% or 7.1% in OE2 or OE11 response to Cd (Figure 2A,B). Likewise, the activity of CAT in OE2 or OE11 was 2.2 or 1.9 times of WT (Figure 2C). In sum, the overexpression of *IlHMA2* could reinforce the antioxidant activity and proline accumulation, which jointly contributed to scavenging excess reactive oxygen species (ROS) and maintaining membrane integrity in tobacco under Cd contamination. 

### 2.3. Overexpression of IlHMA2 Enhanced the Root-to-Shoot Cd Translocation in Tobacco under Cd Exposure

To explore whether the overexpression of *IlHMA2* could improve root-to-shoot Cd translocation, Cd accumulation in WT plants and transgenic lines (OE2 and OE11) were investigated after 50 mg/L Cd treatment for 14 days. Under Cd treatment, Cd concentration in shoots remarkably increased by 22.84% and 17.89%, but Cd concentration in roots decreased by 11.78% and 6.15% in OE2 and OE11 compared to WT plants, respectively (Figure 3A,B). Moreover, a time-course experiment showed that Cd concentration in the root xylem saps was always higher in OE2 and OE11 than in WT plants (Figure 5A). These results resulted in a 1.42- and 1.26-fold increase in TF-Cd in transgenic tobacco relative to WT plants (Figure 3C). Meanwhile, BCF-Cd in transgenic tobaccos was 1.23- and 1.18-times higher than that in WT plants (Figure 3D). These results indicated that overexpression of *IlHMA2* could facilitate the root-to-shoot Cd translocation in tobacco by enhancing the Cd concentration of xylem saps. This would further contribute to improving Cd phytoremediation efficiency. 

### 2.4. Overexpression of IlHMA2 Promoted the Accumulation of Zn in Tobacco under Cd Exposure

We further examined the contribution of the overexpression of *IlHMA2* in enhancing the root-to-shoot translocation of Zn in tobacco under Cd exposure for 14 days. There was no significant difference in Zn concentration of shoots and roots between WT and transgenic lines (OE2 and OE11) without Cd stress (Figure 4A,B). However, Zn concentration in transgenic lines was significantly higher than that in WT under Cd treatment. For example, Zn concentration in shoots or roots increased by 75.4% or 30.73% in OE2 relative to WT (Figure 4A,B). The increase in Zn concentration in shoots and roots was ascribed to the significant increase in the Zn concentration of xylem saps in transgenic lines (Figure 5B). Zn translocation factor in OE2 and OE11 increased by 34.9% and 41.3% under Cd stress, respectively, compared to WT plants (Figure 4C). These findings implied that the overexpression of *IlHMA2* could promote Zn accumulation, which is required for the maintenance of Zn homeostasis in tobacco exposed to Cd contamination.

## 3. Discussion

### 3.1. Overexpression of IlHMA2 Contributed to Scavenging the ROS in Tobacco via Modulating Antioxidant Systems under Cd Contamination

Cd is a non-essential heavy metal that exerts negative effects on plants growth. Once Cd was absorbed and transported into the plants, it induced cellular oxidative stress [20]. For instance, Cd stress caused H_2_O_2_ accumulation, leading to the overproduction of ROS, triggering the oxidative malfunction of DNA and proteins [21]. Moreover, oxidative stress can increase MDA levels in plants, which signifies heightened lipid peroxidation, affecting cellular membranes, lipoproteins, and other lipid-containing molecules [22]. Generally, plants have evolved several adaptive mechanisms to detoxify ROS and minimize oxidative damage [2]. Antioxidant enzymes, such as SOD, POD and CAT, play important roles in detoxifying ROS. This is attributed to the increase in antioxidant enzyme activity, which effectively eliminated ROS, thereby protecting plants from oxidative damage [23]. SOD, an important ROS-scavenging antioxidant enzyme, is involved in blocking O^2−^-driven cell damage via catalyzing the detoxification of O^2−^ to O_2_ [24]. Our present study showed that a higher activity of SOD was found in transgenic tobacco under Cd treatment (Figure 2A). This indicates that the overexpression of *IlHMA2* can increase the activity of the antioxidative system to scavenge O^2−^ accumulation. In addition, POD is mainly responsible for the removal of ROS (H_2_O_2_) [25]. We found the overexpression of *IlHMA2* stimulated the increase in POD activity (Figure 2B), which was beneficial for decreasing the accumulation of H_2_O_2_ and MDA in transgenic tobacco compared with WT plants (Table 1). This demonstrates that the overexpression of *IlHMA2* could lower the level of ROS, minimizing oxidative damage by elevating the activity of POD. On the other hand, POD is an important component for lignin biosynthesis, which acts as a physical barrier against heavy metals [26]. This might be partly contributed to the accumulate lignin to improve Cd tolerance in transgenic tobacco via increasing the activity of POD. However, this still requires examination in further studies. It is known that CAT is involved in converting reactive H_2_O_2_ and O_2_ to benign H_2_O [27]. Cd can also replace Fe in the active site and further inhibit the activity of CAT, causing the overproduction of ROS and thus inducing cell oxidative damage [28,29]. However, *IlHMA2-*expressing tobacco significantly inhibited the decrease in CAT activity relative to WT plants, accompanied by the reduction in the level of H_2_O_2_ and MDA. This illustrates that overexpressing *IlHMA2* might alleviate the CAT dysfunction caused by Cd, further it is helpful to lower the overproduction of ROS and protect the membrane integrity. At the same time, the nonenzymatic antioxidant, proline, accumulated continuously with increased Cd stress to play a long-term role in scavenging ROS [30]. This explains that overexpressing *IlHMA2* is involved in eliminating ROS and reducing MDA contents by enhancing the accumulation of proline in tobacco under Cd exposure. Taken together, the overexpression of *IlHMA2* could contribute to the elimination of the overproduction of ROS and the maintenance of membrane integrity by reinforcing the antioxidative defense system, thereby improving the growth of tobacco under Cd exposure.

### 3.2. Overexpression of IlHMA2 Facilitated the Root-to-Shoot Zn Translocation to Improve the Physiological Zn Status in Tobacco under Cd Contamination

Zn is an essential micronutrient that is required for many biochemical pathways [31]. However, Cd can be transported into plants in competition with Zn for binding sites in plasma membrane, xylem loading, and cell walls due to similar physical and chemical properties [2]. Therefore, it would interfere with the uptake, transport and utilization of Zn inside the cell [4]. In this study, no significant difference was observed in Zn concentration between WT and transgenic tobaccos under normal growth conditions (Figure 4A,B). Despite the Zn concentration being dramatically decreased in both WT and transgenic tobaccos compared to the control, *IlHMA2-*expressing tobaccos enhanced the Zn concentration relative to WT plants under Cd treatment. Previous documents showed that HMA2 could act as a Zn pump responsible for loading Zn into the xylem to long-distance transport to shoots, thus it is required for cytoplasmic Zn homeostasis [10,12,32]. More importantly, the overexpression of *IlHMA2* increased Zn concentration in xylem saps in tobacco in response to Cd compared to WT plants. These reflect that the overexpression of *IlHMA2* could accelerate Zn efflux into the xylem, further enhancing root-to-shoot Zn translocation and thereby improving cellular Zn status and the ability to withstand Cd stress. Moreover, Sinclair et al. (2018) [33] found that HMA2 and MTP2 functioned in the roots of Zn-deficient plants to improve the partitioning of Zn to the shoot. We proposed that the overexpression of *IlHMA2* might improve the physiological Zn status of shoots by synergistically regulating the transcript levels of *MTP2* in tobacco under Cd exposure. Taken together, these pieces of evidence suggest that the overexpression of *IlHMA2* facilitates Zn translocation from roots to shoots via improving Zn efflux pumping into the xylem, thereby maintaining cellular Zn homeostasis and improving its Cd tolerance. 

### 3.3. Overexpression of IlHMA2 Enhances Cd Phytoremediation Efficiency in Tobacco 

Root-to-shoot Cd translocation is a key factor to improve the Cd phytoremediation efficiency [4]. HMA2 has been identified as an important transporter mediating Cd loading into the xylem, which is responsible for long-distance Cd transport to shoots [19]. This indicates that it is closely associated with the efficiency of phytoremediation. Several documents showed that the overexpression of *TaHMA2* enhanced root-to-shoot Cd translocation in rice, but that it increased sensitivity to Cd in transgenic plants [14]. On the contrary, the overexpression of *AtHMA4* not only increased Cd translocation, but also improved its Cd tolerance [15]. We agree with the viewpoint that overexpressing *IlHMA2* could improve the root-to-shoot Cd translocation by enhancing Cd concentrations in xylem sap (Figure 3C and Figure 5A). However, we hold a different opinion on whether *HMA2-*expressing plants have a decrease in Cd resistance. One important reason is that higher concentrations of Zn and lower levels of ROS could jointly alleviate Cd toxicity in transgenic tobacco, thereby improving its tolerance, as mentioned above. And beyond that, we observed that the Cd concentration of roots slightly decreased in transgenic tobacco compared to WT (Figure 3B). It might be due to the fact that the overexpression of *IlHMA2* could increase Cd translocation, further stimulating Cd accumulation in shoots to alleviate this toxic metal in root cells. Our previous study showed that IlHMA2 loaded Cd into the xylem is responsible for long-distance transport to shoots. Subsequently, tonoplast transporters IlHMA3, IlCAX, and IlMTP1 jointly sequestrated Cd into vacuoles to the detoxification and accumulation of Cd [19]. The implication is that the overexpression of *IlHMA2* might indirectly trigger an upregulated expression of *HMA3*, *CAX,* or *MTP1*, further promoting the accumulation of Cd in shoots. This also further explains why the value of bioaccumulation factors is higher in transgenic lines (Figure 3D), thus contributing to improve Cd phytoremediation efficiency.

## 4. Materials and Methods

### 4.1. Plant Material and Growth Condition

Seeds of *I. lactea* were collected from the Beijing Academy of Agriculture and Forestry Sciences. They were sterilized with 5% sodium hypochlorite solution (*v*/*v*) for 5 min and rinsed thoroughly with distilled water, soaked in water for 56 h at 40 °C, then sown in plastic culture pots containing peat and sand (*v*/*v*, 2:1) under 25 °C/18 °C (day/night), the daily photoperiod was 16 h/8 h (light/dark; light intensity was 600 μmol/m^2^/s) and relative humidity (RH) was about 60%. Pots were watered every three days for four weeks. Once plants had three leaves, seedlings were selected for uniformity, then transferred into plastic containers (length 19 cm, width 14 cm and height 8 cm) filled with 0.6 L modified Hoagland solution containing 2mM KNO_3_, 1 mM NH_4_H_2_PO_4_, 0.5 mM Ca (NO_3_)_2_·4H_2_O, 0.5 mM MgSO_4_·7H_2_O, 60 μM Fe-citrate, 92 μM H_3_BO_3_, 18 μM MnCl_2_·4H_2_O, 1.6 μM ZnSO_4_·7H_2_O, 0.6 μM CuSO_4_·5H_2_O, and 0.7 μM (NH_4_)_6_Mo_7_O_24_·4H_2_O for two weeks [2].

### 4.2. Cloning and Vector Construction

Total RNA was extracted from the roots of *I. lactea* using the Trizol kit (TaKaRa, Beijing, China), then the first strand cDNA was synthesized from 1 μg of total RNA using a 5×Primescript RTase mix (Takara, Beijing, China). The open reading frame (ORF) of *IlHMA2* (GenBank accession: KY696282) was amplified by the primers P1 (5′-ATGGAATTCTTCCAAGTACTA-3′) and P2 (5′-TTATTCGCTTCCACTCCCATG-3′). The PCR products were purified from agarose gels and cloned into pMD-19T cloning vector followed by sequencing (Sangon, Shanghai, China). Then, the ORF of *IlHMA2* with P3 (5′-CATGGTAGATCTATGGAATTCTTCCAAGTACTA-3′) and P4 (5′-ATTCGAGCTGGTCACCTTATTCGCTTCCACTCCCATG-3′) was amplified and cloned into the *Bgl II*/*BstE II* site of the pCAMBIA3301 vector to construct the 35S:IlHMA2 plasmid using infusion cloning technology.

### 4.3. Plant Transformation and Molecular Characterization

The experiment was performed using tobacco (*Nicotiana tabacum* L. cv. Wisconsin 38). The plasmid was transformed into tobacco plants according to the *Agrobacterium*-mediated leaf disc transformation method [34]. Firstly, 35S:IlHMA2 plasmid was transformed into *Agrobacterium tumefaciens* EHA105, then cultured in YEB medium (including 50 mg/L Kan and 50 mg/L Rif) at 28 °C. A single colony was selected and cultivated to OD_600_ of 0.4 in liquid medium for infecting tobacco leaves. Secondly, they were cultivated on Murashige and Skoog (MS) medium containing 2 mg/L 6-benzylaminopurine (6-BA) and 0.2 mg/L 1-naphthaleneacetic acid (NAA) and with pH 5.8 for 3 d. Thirdly, the infected explants were placed into the selection MS medium containing 6-BA and NAA, 8 mg of L-1 glufosinate ammonium (GLA) and 500 mg/L cefotaxime (Cef) and grown in a greenhouse at a 16 h photoperiod, an irradiance of 600 μmol/m^2^/s, a temperature of 25 °C, and a relative humidity of 60% for 4 weeks to induce shoot development. Meanwhile, we will aseptically excise 3–5 cm long shoots and transfer them into rooting 1/2 MS medium supplemented with 0.05 mg/L indole-3-butyric acid, 8 mg/L GLA, and 250 mg/L Cef (pH 5.8), and independent transgenic lines were obtained. They were transplanted into a pot with vermiculite/nutrient soil (1:1) to finally obtain T_2_ transgenic tobacco plants.

Genomic DNA was extracted from young leaves of WT and T2 transgenic tobaccos using a MiniBEST universal genomic DNA extraction kit (TaKaRa, Beijing, China). Positive transgenic plants were verified by the specific primers P5 (5′-TCGTCTTCTCGGTGGTAACTAA-3′) and P6 (5′-AACAAGATGACTCGCAGGATTT-3′). As mentioned above, total RNA was extracted using a Trizol kit following the manufacturer’s instructions, then cDNA was obtained using a PrimeScript RT Reagent Kit (TaKaRa, Beijing, China). Subsequently, the relative expression levels of *IlHMA2* in transgenic tobacco lines were validated by reverse transcription reverse transcription PCR (sqRT-PCR) using the primers pairs P5 and P6. *NtACTIN* was used for the internal control in sqRT-PCR. The fragment of *NtACTIN* was amplified by primer pairs A1 (5′-TATGCTAGTGGTCGTACAACTG-3′) and A2 (5′-AACAACCTTAATCTTCATGCTG-3′). The OE2 and OE11 lines were randomly selected for further analysis.

### 4.4. Cd treatments and Measurement of Growth Parameters for Transgenic Lines

WT and T2 transgenic tobaccos (OE2 and OE11 lines) were grown in greenhouse. They were sown in the vermiculite/nutrient (1:1) soil under ambient conditions (photoperiod: 16h/8h (day/night); temperature: 25 °C/18 °C (day/night); relative humidity: 60%). After growth for 8 weeks, the WT, OE2 and OE11 lines were treated with 0 and 50 mg/L CdCl_2_·2.5H_2_O. After 14 days treatments, the WT plants were harvested, and the plant height and root length were measured for the OE2 and OE11 lines, respectively. Then, the plants were placed in the oven at 80 °C for constant weight, and the dry weight of shoots and roots were measured. Moreover, they were further selected for measuring Cd in the xylem saps of roots according to the method of Guo et al. (2019) [19]. Every treatment was replicated eight times.

### 4.5. Measurement of H_2_O_2_, MDA, and Antioxidant Enzymes for Transgenic Lines

The level of H_2_O_2_ was measured [35]. Firstly, 0.1g fresh leaves was weighed out then ground until homogenization, after which 1 mL acetone was added, the content was centrifuged at 8000× *g* at 4 °C, the supernatant was collected, and 5% titanium sulfate solution and ammonia water was added. The mixture was centrifuged at 4000× *g* for 10 min, then pellets with 2 mM sulfuric acid were dissolved, and the absorbance at 415 nm was measured. To evaluate lipid peroxidation, malondialdehyde (MDA) content was measured [2]. The absorbance at 450 nm, 532 nm, and 600 nm was measured for the calculation of MDA. Antioxidant enzyme activity, such as SOD (EC 1.15.1.1), POD (EC 1.11.1.7), and CAT (EC 1.11.1.6), were all measured for the evaluation of antioxidant ability. The activity of SOD, POD, and CAT was measured [36]. The activity of SOD was calculated by measuring the absorbance at 560 nm, and the POD activity was measured at 470 nm. The activity of CAT was measured at 405 nm with the method of ammonium molybdate colorimetry. Proline was also an important index of stress resistance. The content of proline was measured by ninhydrin colorimetry, and calculated by the absorbance at 520 nm [37]. 

### 4.6. Determination of Zn and Cd Concentration

At the end of the experiment, the plants were harvested and divided into shoots and roots. The dried shoots and roots were ground into power. The powers were dissolved in a mixture of HNO_3_ and HCLO_4_ (4:1), then digested by microwave acid digestion until near dryness. The residue was dissolved with distilled water to a total volume of 25 mL, and the solution was filtered with a 0.45 μm filter. Zn and Cd concentration was measured using an atomic absorption spectrophotometer (AA-6300C, Shimadza, Kyoto, Japan). Translocation factors (TF) or bioaccumulation factors (BCF) were calculated as follows [2]: TF= (Cd/Zn in shoots)/(Cd/Zn in roots) reflects a plant’s ability to translocate Cd/Zn from roots to shoots; and BCF= (Cd in shoots)/(Cd in solution) indicates the ability of plants to accumulate Cd.

### 4.7. Statistic Analysis 

Data were presented as means with standard deviation (SD). One-way analysis of Variance and Duncan’s multiple range tests were performed using SPSS 22.0 (IBM, Armonk, NY, USA). The graph was drawn with Origin 9.5 (OriginLab, Northampton, MA, USA).

## 5. Conclusions

Our findings suggest that the overexpression of *IlHMA2* confers increased root-to-shoot Cd translocation via elevating Cd concentrations of xylem saps. Moreover, the overexpression of *IlHMA2* could facilitate Zn accumulations, accompanied by increasing proline content and antioxidant enzyme activity to lower the overproduction of ROS in transgenic tobacco exposed to Cd. These contributed to alleviating the inhibitory effect of Cd toxicity on transgenic tobacco, further improving the growth and tolerance. Therefore, the overexpression of *IlHMA2* is a promising tool for improving Cd phytoremediation efficiency.

## Figures and Tables

**Figure 1 plants-12-03460-f001:**
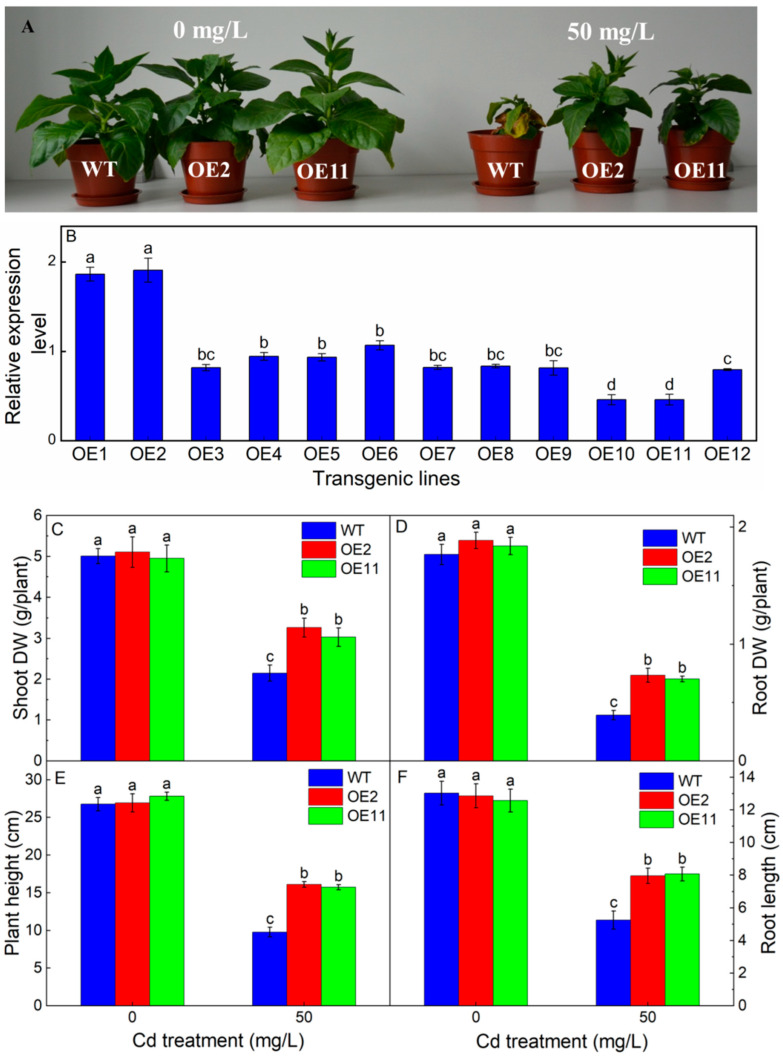
The growth phenotype (**A**), shoot dry weight (**B**), relative expression level of *IlHMA2* in leaves of transgenic tobaccos by sqRT-PCR (**C**), root dry weight (**D**), plant height (**E**), and root length (**F**) of wild-type (WT) and transgenic tobacco lines (OE2, OE11) under the control (0 mg/L CdCl_2_·2.5H_2_O) and 50 mg/L CdCl_2_·2.5H_2_O exposure for 14 days. Data are means ± SD (*n* = 8) and bars indicate SD. Different letters indicate significant differences at *p* < 0.05 (Duncan’s test).

**Figure 2 plants-12-03460-f002:**
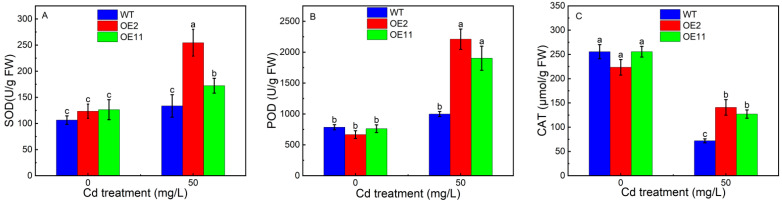
Response of antioxidant enzyme activity SOD (**A**), POD (**B**), CAT (**C**) in wild-type (WT) and transgenic tobacco lines (OE2, OE11) under the control (0 mg/L CdCl_2_·2.5H_2_O) and 50 mg/L CdCl_2_·2.5H_2_O exposure for 14 days. Data are means ± SD (*n* = 8) and bars indicate SD. Different letters indicate significant differences at *p* < 0.05 (Duncan’s test).

**Figure 3 plants-12-03460-f003:**
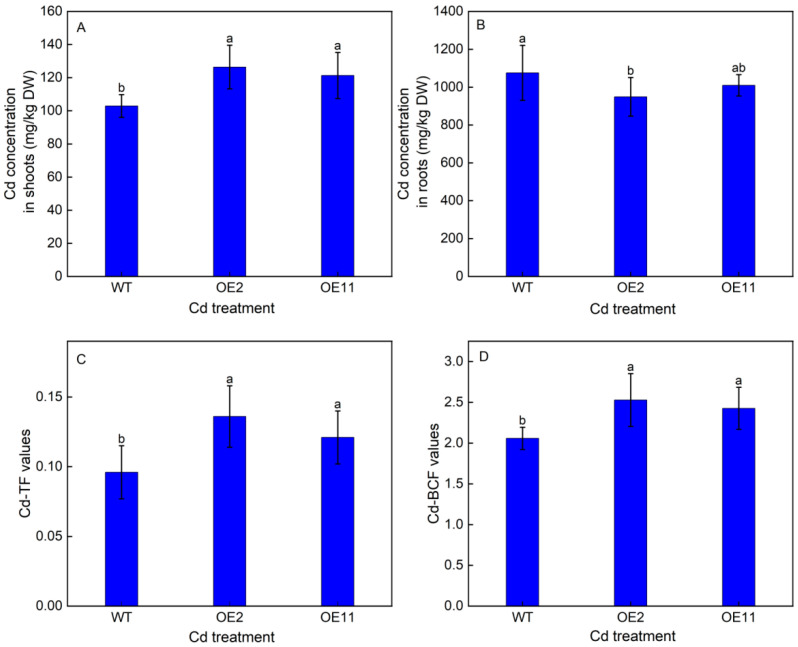
Cd concentrations in shoots (**A**) and roots (**B**), and TF-Cd (**C**) and BCF-Cd (**D**) values of wild-type (WT) and transgenic tobacco lines (OE2, OE11) under the control (0 mg/L CdCl_2_·2.5H_2_O) and 50 mg/L CdCl_2_·2.5H_2_O exposure for 14 days. TF reflects a plant’s ability to translocate Cd from roots to shoots, BCF indicates the ability of plants to accumulate Cd. Data are means ± SD (*n* = 8) and bars indicate SD. Different letters indicate significant differences at *p* < 0.05 (Duncan’s test).

**Figure 4 plants-12-03460-f004:**
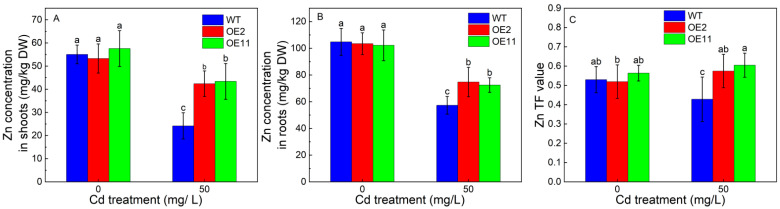
Zn concentrations in shoots (**A**) and roots (**B**), TF-Zn (**C**), and values of wild-type (WT) and transgenic tobacco lines (OE2, OE11) under the control (0 mg/L CdCl_2_·2.5H_2_O) and 50 mg/L CdCl_2_·2.5H_2_O exposure for 14 days. TF reflects a plant’s ability to translocate Zn from roots to shoots. Data are means ± SD (*n* = 8) and bars indicate SD. Different letters indicate significant differences at *p* < 0.05 (Duncan’s test).

**Figure 5 plants-12-03460-f005:**
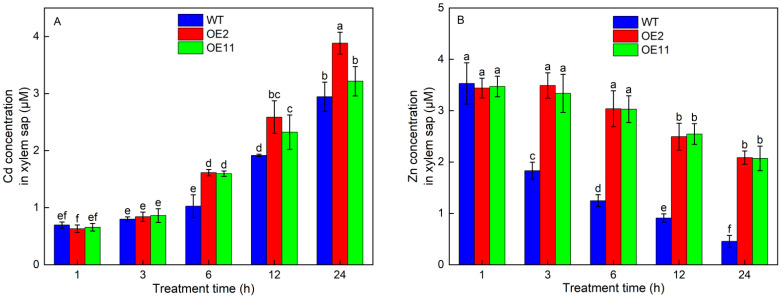
Time-dependent change of Cd (**A**) and Zn (**B**) concentrations in root xylem saps of wild-type (WT) and transgenic tobacco lines (OE2, OE11) under 50 mg/L CdCl_2_·2.5H_2_O exposure for 1 h, 3 h, 6 h, 12 h, 24 h. Data are means ± SD (*n* = 8) and bars indicate SD. Different letters indicate significant differences at *p* < 0.05 (Duncan’s test).

**Table 1 plants-12-03460-t001:** Changes in H_2_O_2_, MDA, and proline content in WT and transgenic tobacco under the control (0 mg/L CdCl_2_·2.5H_2_O) and 50 mg/L CdCl_2_·2.5H_2_O exposure for 14 days. Values are means ± SD (*n* = 8). Different letters indicate significant differences at *p* < 0.05 (Duncan’s test).

Cd Treatment (mg/L)	Lines	MDA (mmol/g FW)	H_2_O_2_ (μmol/g FW)	Proline (μg/g FW)
0	WT	0.31 ± 0.03 c	2.05 ± 0.07 b	0.69 ± 0.02 b
	OE2	0.31 ± 0.04 c	2.16 ± 0.23 b	0.74 ± 0.05 b
	OE11	0.28 ± 0.03 c	1.95 ± 0.27 b	0.71 ± 0.07 b
50	WT	1.3 ± 0.06 a	3.82 ± 0.46 a	0.75 ± 0.02 b
	OE2	0.74 ± 0.09 b	2.52 ± 0.58 b	1.53 ± 0.17 a
	OE11	0.73 ± 0.05 b	2.67 ± 0.57 b	1.61 ± 0.33 a

## Data Availability

The data generated and analyzed during this study are included in this article.

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
