# Peer review of "Overexpression of *IlHMA2*, from *Iris lactea*, Improves the Accumulation of and Tolerance to Cadmium in Tobacco"

_plants, 2023, doi:10.3390/plants12193460_

Round 1
Reviewer 1 Report
1. Figure 2 the activity of antioxidant enzymes SOD (A), POD (B), and CAT (C) was measured in wild-type (WT) and transgenic tobacco lines (OE2, OE11) before and after exposure to 0 mg/L and 50 mg/L CdCl2·2.5 H2O for 14 days. Why is the CAT content lower after treatment?
2. In Figure 3-4, please explain what TF/BCF stand for respectively? TF= (Cd/Zn in shoots)/(Cd/Zn in roots) reflects a plant's ability to translocate Cd/Mn from roots to shoots?
3. Can the functional mechanism of IlHMA2 transport be elucidated through heterologous expression or knockout mutation?
The language of the manuscript should be improved.
Author Response
Dear Editors and Reviewers,
We thanks for your time and patience, as well as your constructive and thoughtful comments. We have addressed all comments in the revised manuscript, and responded point by point to the comments as itemized below. If the manuscript needs further revision, please tell us.
Reviewers’ comments:
- Figure 2 the activity of antioxidant enzymes SOD (A), POD (B), and CAT (C) was measured in wild-type (WT) and transgenic tobacco lines (OE2, OE11) before and after exposure to 0 mg/L and 50 mg/L CdCl22.5 H2O for 14 days. Why is the CAT content lower after treatment?
Answer:Thank you for your comments. The enzyme activity of CAT decreased under Cd treatment compared with control in this study. Previous study showed that Cd pressure can inhibit the uptake level of nutrients such as Fe and Zn, and then further decrease the activity of antioxidant enzyme (Gill et al. 2010). For example, the CAT activity was found to be decreased in Sorghum bicolor under Cd treatment (Hassan et al. 2020). This is the similar with our results. Although the activity of CAT was decreased under Cd treatment compared with control, overexpressing IlHMA2 always increased the CAT activity under Cd treatment compared with WT plants. This could be contributed to scavenging the H2O2.
Gill, S.S.; Tuteja, N. Reactive oxygen species and antioxidant machinery in abiotic stress tolerance in crop plants. Plant Physiol. Bioch. 2010, 48, 909-930.
Hassan, M.J.; Raza, M.A.; Rehman, S.U.; Ansar, M.; Gitari, H.; Khan, I. ; Wajid, M.; Ahmed, M.; Shah G.S.; Peng, Y.; Li, Z. Effect of cadmium toxicity on growth, oxidative damage, antioxidant defense system and cadmium accumulation in two sorghum cultivars. Plants. 2020, 9, 1575.
- In Figure 3-4, please explain what TF/BCF stand for respectively? TF= (Cd/Zn in shoots)/(Cd/Zn in roots) reflects a plant's ability to translocate Cd/Mn from roots to shoots?
Answer: Thank you. We added the explanation of TF and BCF in Figure 3 and 4 as follows, according to your suggestions.
Figure 3. Cd concentrations in shoots (A) and roots (B), TF-Cd (C) and BCF-Cd (D) values of wild-type (WT) and transgenic tobacco lines (OE2, OE11) under the control (0 mg/L CdCl2•2.5H2O) and 50 mg/L CdCl2•2.5H2O exposure for 14 days. TF reflects a plant's ability to translocate Cd from roots to shoots, BCF indicates the ability of plants to accumulate Cd. Data are means ± SD (n = 8) and bars indicate SD. Different letters indicate significant differences at P < 0.05 (Duncan’s test).
Figure 4. Zn concentrations in shoots (A) and roots (B), TF- Zn (C) and BCF- Zn (D) values of wild-type (WT) and transgenic tobacco lines (OE2, OE11) under the control (0 mg/L CdCl2•2.5H2O) and 50 mg/L CdCl2•2.5H2O exposure for 14 days. TF reflects a plant's ability to translocate Zn from roots to shoots. Data are means ± SD (n = 8) and bars indicate SD. Different letters indicate signifi-cant differences at P < 0.05 (Duncan’s test).
- Can the functional mechanism of IlHMA2 transport be elucidated through heterologous expression or knockout mutation?
Answer: Thank you for your valuable comments. The document demonstrated that HMA2 can enhance root-shoot Zn/Cd translocation, but the Cd tolerance was decreased (Tan et al., 2013). Interestingly, Verret et al. (2004) reported that overexpressing AtHMA4 not only improved root-to-shoot Cd translocation, but also conferred the increased Cd tolerance (Verret et al., 2004). Our previous study showed that IlHMA2 loading Cd into the xylem was involved in long-distance transport to shoots (Guo et al., 2019). However, the contribution of overexpressing IlHMA2 to enhance root-to-shoot Cd translocation and improve Cd tolerance in transgenic plant is still unexplored till. In view of this, we overexpressed IlHMA2 in tobacco, results suggested that overexpression of IlHMA2 could facilitate Cd accumulation via enhancing root-to-shoot Cd translocation and conferred increased Cd tolerance in tobacco. This would provide a valuable reference for improving Cd phytoremediation efficiency.
References
- Guo, Q.; Tian, X.X.; Mao, P.C.; Meng, L. Functional characterization of IlHMA2, a P1B2-ATPase in Iris lactea response to Cd. Exp. Bot. 2019, 157, 131–139.
- Tan, J.; Wang, J.; Chai, T.; Zhang, Y.X.; Feng, S.S.; Li, Y.; Zhao, H.J.; Liu, H.M.; Chai, X.P. Functional analyses of TaHMA2, a P1B -type ATPase in wheat. Plant Biotechnol. J. 2013, 11, 420–431.
- Verret, F.; Gravot, A.; Auroy, P.; Leonhardt, N.; David, P.; Nussaume, L.; Vavasseur, A.; Richaud, P. Overexpression of AtHMA4 enhances root-to-shoot translocation of zinc and cadmium and plant metal tolerance. FEBS Lett. 2004, 576, 306–312.

Reviewer 2 Report
1. I consider it appropriate to add some references of plant tolerance to cadmium to the Introduction (e.g.DOI: 10.1016/j.envexpbot.2020.104327)
2.
4.4. Cd treatments and measurement of growth parameters for transgenic lines
“WT, OE2 and OE11 lines were treated with 0 and 50 mg/L CdCl2·2.5H2O.” How many CdCl2·2.5H2O was used to water the plants?
3.
“IlACTIN was used for the internal control in sqRT-PCR. The fragment of IlACTIN was amplified by primers pairs A1 (5’-TATGCTAGTGGTCGTACAACTG-3’) and A2 (5’-AACAACCTTAATCTTCATGCTG-3’).”?? Figs1, Relative expression level of IlHMA2 should be relative to tobacco actin gene.
4. It‘s better to use three transgenic lines in all experiments.
Minor editing of English language required
Author Response
Dear Editors and Reviewers,
We thanks for your time and patience, as well as your constructive and thoughtful comments. We have addressed all comments in the revised manuscript, and responded point by point to the comments as itemized below. If the manuscript needs further revision, please tell us.
Reviewers’ comments:
- I consider it appropriate to add some references of plant tolerance to cadmium to the Introduction (e.g.DOI: 10.1016/j.envexpbot.2020.104327)
Answer: Thank you for your precious comments. We carefully read this reference and added it to the introduction section as follows.
Notably, overexpressing stress response membrane protein genes, e.g.OsSMP1, conferred increased plants tolerance to Cd stress [17].
[17] Zheng, S.W.; Liu, S.B.; Feng, J.H.; Wang, W.; Wang, Y.W.; Yu, Q.; Liao, Y.R.; Mo, Y.H.; Xu, Z.J.; Li, L.H.; Gao, X.L.; Jia, X.M.; Zhu, J.Q.; Chen, R.J. Overexpression of a stress response membrane protein gene OsSMP1 enhances rice tolerance to salt, cold and heavy metal stress. Environ. Exp. Bot. 2021,182, 104327.
- Cd treatments and measurement of growth parameters for transgenic lines
“WT, OE2 and OE11 lines were treated with 0 and 50 mg/L CdCl2·2.5H2O.” How many CdCl2·2.5H2O was used to water the plants?
Answer: Thanks for your comments. We watered 500 mL solution of 50 mg/L CdCl2•2.5H2O to the soil all at once. Then watered the plants with Hoagland solution every four days.
- “IlACTIN was used for the internal control in sqRT-PCR. The fragment of IlACTIN was amplified by primers pairs A1 (5’-TATGCTAGTGGTCGTACAACTG-3’) and A2 (5’-AACAACCTTAATCTTCATGCTG-3’).”?? Figs1, Relative expression level of IlHMA2 should be relative to tobacco actin gene.
Answer: We sincerely apologize for this error. Primer pairs A1 (5’-TATGCTAGTGGTCGTACAACTG-3’) and A2 (5’-AACAACCTTAATCTTCATGCTG-3’) can be used to amplify the fragment of NtACTIN. We revised it in the manuscript.
- It‘s better to use three transgenic lines in all experiments.
Answer: Thank you for your valuable comments. In this study, we screened 12 positive lines of transgenic tobaccos. And then measured the relative expression level of IlHMA2 by sqRT-PCR. The relative expression level was among 0.46-1.91 times in twelve lines. So, we chosen the highest expression level (OE2) and the lowest expression level (OE11) of IlHMA2 in twelve transgenic tobaccos for further study. We appreciate your suggestions, and we will use three lines in our future study.

Reviewer 3 Report
This study investigated the question whether overexpressing IlHMA2, a plasma membrane transporter-encoding gene isolated from Iris lacteal, would enhance Cd tolerance, and improve its phytoremediation. Thus, authors generated 12 independent transgenic tobacco IlHMA2 OE (overexpression) plants and investigated the translocation and accumulation of Cd and Zn (known to have an antagonistic relationship with Cd from previous studies). They also quantified several widely used biochemical-physiological parameters in the transgenic OE plants relative to WT non-transgenic controls under both regular and Cd stress conditions. They found higher concentrations of Zn and lower level of ROS could alleviate Cd toxicity for transgenic tobacco, thereby improving the growth and Cd tolerance seen in the OE lines compared with the WT plants under the Cd stress conditions.
Overall, I consider results of this study are interesting and important. Conclusions are supported by the data. However, a major problem of this manuscript is English that must be improved thoroughly be for can be considered for acceptance. Also have some additional comments listed below.
1. The figure showing gene expression levels of the 12 independent transgenic lines can be placed in the main figure of the manuscript (Figure1B).
2. Where is Figure 4D?
3. Under Cd treatment, Zn can be absorbed in a large amount in the root, and Zn enhanced the root-to-shoot translocation with Cd from Figure 4? Please check or clarify.
4. The description of Figure 5 is not clear, and the materials will be collected under 50 mg/L CdCl2·2.5H2O exposed for 14 days?
English must be extensively improved.
Author Response
Dear Editors and Reviewers,
We thanks for your time and patience, as well as your constructive and thoughtful comments. We have addressed all comments in the revised manuscript, and responded point by point to the comments as itemized below. If the manuscript needs further revision, please tell us.
Reviewers’ comments:
This study investigated the question whether overexpressing IlHMA2, a plasma membrane transporter-encoding gene isolated from Iris lacteal, would enhance Cd tolerance, and improve its phytoremediation. Thus, authors generated 12 independent transgenic tobacco IlHMA2 OE (overexpression) plants and investigated the translocation and accumulation of Cd and Zn (known to have an antagonistic relationship with Cd from previous studies). They also quantified several widely used biochemical-physiological parameters in the transgenic OE plants relative to WT non-transgenic controls under both regular and Cd stress conditions. They found higher concentrations of Zn and lower level of ROS could alleviate Cd toxicity for transgenic tobacco, thereby improving the growth and Cd tolerance seen in the OE lines compared with the WT plants under the Cd stress conditions.
Overall, I consider results of this study are interesting and important. Conclusions are supported by the data. However, a major problem of this manuscript is English that must be improved thoroughly be for can be considered for acceptance. Also have some additional comments listed below.
- The figure showing gene expression levels of the 12 independent transgenic lines can be placed in the main figure of the manuscript (Figure1B).
Answer:Thank you for your valuable suggestions. We added the gene expression data to the Figure1B as follows.
- Where is Figure 4D?
Answer:Thank you for your comments. We are very sorry for the mistake. Figure 4 has only A, B, and C. There was no D. We had revised and deleted the description of Figure 4D in the revision text version.
- Under Cd treatment, Zn can be absorbed in a large amount in the root, and Zn enhanced the root-to-shoot translocation with Cd from Figure 4? Please check or clarify.
Answer:Thank you for your constructive comments. We explained it in the manuscript in Result 2.4 as follows:
Zn concentration in shoots and roots of transgenic tobacco was significantly higher than that of WT under Cd treatment. For example, Zn concentration in shoots or roots increased by 75.4% or 30.73% in OE2 relative to WT (Figure 4A, B). The increase of Zn concentration in shoots and roots was ascribed to the marked increase in Zn concentration of xylem sap in OE2 and OE11 (Figure 5 B). Zn translocation factor in OE2 and OE11 increased by 34.9% and 41.3% under Cd stress, respectively, compared with WT plants (Figure 4C). These findings imply that overexpressing IlHMA2 could promote Zn accumulation via enhancing root-to-shoot Zn translocation, which is also required for the maintenance of Zn homeostasis in tobacco exposed to Cd.
- The description of Figure 5 is not clear, and the materials will be collected under 50 mg/L CdCl2·2.5H2O exposed for 14 days?
Answer: Thank you. We carefully corrected the description of Figure 5 in the revision version. The details are as follows.
Figure 5. Time-dependent change of Cd (A) and Zn (B) concentrations in root xylem sap of wild-type (WT) and transgenic tobacco lines (OE2, OE11) under 50 mg/L CdCl2·2.5H2O exposure for 1h, 3h, 6h, 12h, 24h. Data are means ± SD (n = 6) and bars indicate SD. Different letters indicate significant differences at P < 0.05 (Duncan’s test).

Round 2
Reviewer 1 Report
The language of the manuscript should be improved.
The language of the manuscript should be improved.
Author Response
Dear Reviewer,
We appreciate your time and patience, as well as your constructive and thoughtful comments. We have addressed all comments in the revised manuscript, and responded to these point by point. All authors agreed on the contents of the paper and post no conflicting interest. We are happy to address any further revisions that are needed.
Reviewer’ comments:
The language of the manuscript should be improved.
Answer: We thank your good suggestions. We invited a native English colleague to polish the language in whole text. All revised places were marked with red font. More modification details, please see the revision version.
